

**Gross primary productivity responses to meteorological drivers: insights from observations**
**and multi-model ensembles**
Yuxin Zheng [1], Xu Yue [1*], Xiaofei Lu [1], Jun Zhu [1]
[1] Jiangsu Key Laboratory of Atmospheric Environment Monitoring and Pollution Control,
Collaborative Innovation Center of Atmospheric Environment and Equipment Technology, School
of Environmental Science and Engineering, Nanjing University of Information Science &
Technology (NUIST), Nanjing, 210044, China
[*] Corresponding authors: Xu Yue (yuexu@nuist.edu.cn)
**Abstract**
Climate change has a substantial impact on ecosystem gross primary productivity (GPP), but
the specific roles of different meteorological factors across various vegetation types remain
unclear. This study investigates GPP responses to variations in temperature, precipitation, and
drought, using data from three observational products and 17 dynamic vegetation models.
Observed GPP showed a positive response to temperature in boreal regions, with sensitivities
ranging from 0.01 to 0.05 g C m$^2$ day$^{-1}$ K$^{-1}$. In contrast, GPP responded negatively to temperature
in the tropics, with sensitivities of -0.07±0.15 g C m$^2$ day$^{-1}$ K$^{-1}$ for evergreen broadleaf forests and
-0.25 ± 0.11 g C m$^2$ day$^{-1}$ K$^{-1}$ for $C_4$ grasslands. Precipitation had a relatively low impact on GPP
in deciduous and evergreen forests, while non-tree species, such as grasslands and croplands,
showed a positive response. GPP sensitivity to drought index (scPDSI) was similar to that of
precipitation, except that observed GPP in evergreen forests negatively responded to scPDSI. The
models generally reproduced these observed patterns but tended to overestimate the effect of
precipitation on GPP. As a result, they predicted higher sensitivity in tropical grasslands to drought
stress but lower resilience in trees. Both observations and simulations exhibited negative GPP
responses to extreme warming and drought on a global scale, though models tended to
overestimate the magnitude of these negative effects. This study distinguished GPP responses to
key meteorological factors across vegetation types and numerical models, providing critical
insights for improving the prediction of terrestrial carbon sinks and promoting the climatic
resilience of ecosystems.

**Keywords:** Gross primary productivity; multi-model ensemble; sensitivity; meteorology; drought



**1 Introduction**

Gross primary productivity (GPP) is a critical metric of ecosystem's ability to capture and convert atmospheric carbon dioxide into biomass through photosynthesis, providing the primary carbon input for ecosystems (Fernandez-Martinez et al., 2017; Pinker et al., 2010; Xiao et al., 2019). Therefore, GPP drives key ecosystem processes such as respiration and vegetation growth, which are the foundation of the global carbon cycle (Beer et al., 2010). A variety of environmental and climatic stressors, such as warming and drought, exert a substantial influence on GPP (Tian et al., 2021; Zhang et al., 2022; Liu et al., 2014). Understanding these interactions is vital for predicting ecosystem responses to ongoing and future climate changes, as well as for managing carbon sequestration efforts across diverse biomes.

Climatic factors substantially influence GPP, especially for temperature and precipitation. Temperature affects plant enzymatic activity, generally increasing photosynthetic rates with rising temperatures (Tang et al., 2022). However, this relationship is not linear and can vary across regions. For instance, multi-model simulations show a negative sensitivity of GPP to temperature in tropical regions, with an average sensitivity of $-2.2\pm1.2$ Pg C $yr^{-1}$ $K^{-1}$ (Piao et al., 2013). Precipitation is also a major determinant, affecting water availability and stomatal behavior in plants. Insufficient precipitation may enhance water stress, leading to stomatal closure and reduced photosynthetic rates, thereby decreasing GPP (Liang et al., 2024a). The GPP responses to these climatic factors also vary by plant species. For instance, coniferous forests are more sensitive to temperature and soil moisture, whereas broad-leaved forests are more affected by light and soil nitrogen content (Feng et al., 2007).

As a compound weather extreme, drought typically has negative impacts on GPP by imposing water stress. During drought, plants often close their stomata to conserve water, which reduces $CO_2$ uptake and slows photosynthesis (Granier et al., 2007). While this response is protective in the short term, it ultimately leads to a decline in GPP. Furthermore, reduced soil moisture under drought conditions limits water availability for plant roots, exacerbating water stress and further impairing photosynthesis (Chen and Dominguez, 2024; Brunner and Chartier‐Rescan, 2024). Over the past three decades, there has been a notable increase in the sensitivity of global vegetation productivity to drought conditions. For example, Wei et al. (2023) found that the sensitivity of GPP to drought rose by 13.76% in 2006-2018 compared to 1993-2005. This growing sensitivity indicates that, as droughts become more frequent and severe with climate change (Zscheischler et al., 2014), their impacts on GPP could significantly alter the global carbon cycle, potentially reducing the ability of ecosystems to function as carbon sinks.

Vegetation models are essential for estimating GPP and predicting its future trends under different climate scenarios. These models simulate key processes like vegetation growth, photosynthesis, and carbon cycling, providing valuable insights into how GPP may respond to environmental changes (Piao et al., 2013; Yue et al., 2024). However, large uncertainties arise due to differences in model structures, algorithms, and parameter settings (Poulter et al., 2014; Zhang et al., 2018). To address these, the multi-model ensemble approach has emerged as a promising strategy. For example, the Global Carbon Project (GCP) utilized the ensemble mean of 16 models to estimate land carbon sinks over the past century (Friedlingstein et al., 2022). By integrating models with different structures and parameterizations, this approach reduces model-specific biases and enhances the accuracy of ecosystem simulations. Nevertheless, it can also amplify common biases across models. For instance, Piao et al. (2013) found that process-based models



tend to overestimate the effect of precipitation on GPP while underestimate the response of net
primary productivity (NPP) to temperature changes. This underscores the need of careful
calibration and validation using observed data to improve model reliability.
In this study, we aim to address these challenges by evaluating GPP responses to key climatic
factors using a multi-model ensemble approach. We analyze outputs from 17 vegetation models
participating in the TRENDY project (Sitch et al., 2024) in support of the GCP. We focus on how
well these models reproduce observed relationships between GPP and climatic variables such as
temperature, precipitation, and drought. In addition, we compare the simulated GPP sensitivities
across models and plant functional types (PFTs). We seek to enhance our understanding of the
complex interactions between climatic factors and GPP, ultimately providing more reliable
projections of ecosystem responses to future climate change.

**2 Methods and data**
2.1 Observations of GPP
This study validates models using three GPP observational datasets. The Global Land Surface
Satellite (GLASS) dataset provides a global GPP product from 1981 to 2017 at a spatial resolution
of 0.05º (Liang et al., 2013). It integrates data from the Moderate Resolution Imaging
Spectroradiometer (MODIS) and Advanced Very High Resolution Radiometer (AVHRR), using
optimized light-use efficiency models to estimate GPP by combining absorbed PAR with
environmental factors such as leaf area index and shortwave radiation (Liang et al., 2024b). The
GLASS dataset has been validated against ground measurements and aligns well with observed
GPP, capturing seasonal and interannual variability across ecosystems (Ma and Liang, 2022).
The Global Ozone Monitoring Experiment Solar-Induced Fluorescence (GOSIF) dataset
provides global GPP derived from solar-induced chlorophyll fluorescence (SIF), a proxy for
photosynthetic activity. This dataset is produced using machine learning approach that integrates
discrete SIF measurements from the Orbiting Carbon Observatory-2 (OCO-2) with continuous
spatial and temporal data from the Enhanced Vegetation Index (EVI) obtained from MODIS, and
meteorological reanalysis data from Modern-Era Retrospective analysis for Research and
Applications, Version 2 (MERRA-2) (Li and Xiao, 2019). GOSIF provides GPP data at 0.05º
spatial resolution and 8-day intervals from 2001 to 2018.
The third dataset, short as JUNG, is a benchmark global product that estimates GPP using the
Model Tree Ensemble algorithm (Jung et al., 2011). This dataset integrates satellite vegetation
indices and flux measurements from the FLUXNET network, providing global coverage at 0.5º
spatial resolution and spanning from 1982 to 2011. The JUNG GPP product has been extensively
validated against flux tower measurements, ensuring accurate spatiotemporal GPP estimates (Jung
et al., 2009). It has been widely used in climate and carbon cycle modeling, offering valuable
insights into the impacts of climate change on global vegetation productivity (Wu et al., 2021;
Harper et al., 2018; Leng et al., 2024).

2.2 Model data from the TRENDY project
TRENDY, which stands for "Trends and drivers of the regional scale terrestrial sources and
sinks of carbon dioxide", investigates changes in land carbon fluxes using a multi-model ensemble
approach (Le Quéré et al., 2018; Zeng et al., 2022). For 2022, the project (TRENDY v-11)



involves a total of 17 Dynamic Global Vegetation Models (DGVMs, Table 1), all driven by the
same forcing data including $CO_2$ concentrations, meteorological reanalyses, and land use change
(LUC). TRENDY examines various aspects such as model structures, parameter settings, input
data quality, and the consistency of simulation results. The models follow standardized
experimental protocols to isolate the effects of $CO_2$ fertilization, climate change, and LUC on
global carbon cycle variations (Sitch et al., 2024). Four sets of simulations were performed for
each model with different combinations of time-varying forcings. The S0 run serves as a control
simulation with all forcings fixed to pre-industrial period. The S1 run is similar to S0 but applies
observed $CO_2$ concentrations. The S2 run builds on S1 by incorporating year-to-year variations in
meteorological forcing. The S3 run allows all forcings ($CO_2$, climate, and land use) to evolve over
time, reflecting the real-world environmental changes. For this study, we analyzed simulated GPP
data from the 17 DGVMs for the S2 and S3 experiments, focusing on GPP responses to changes in
major meteorological variables.
2.3 Drought index and meteorological data
The self-calibrating Palmer Drought Severity Index (scPDSI) is an improved version of the
original PDSI and is widely recognized for its application in drought monitoring and assessment
(Wells et al., 2004). The scPDSI enhances the traditional PDSI by incorporating a self-calibration
mechanism that evaluates and adjusts based on past drought events, improving both the accuracy
and stability of the index. Like the original PDSI, the scPDSI relies on time series of precipitation
and temperature, along with location-specific parameters related to soil and surface properties.
However, it introduces updated calculation methods and utilizes more extensive historical
meteorological data to establish baseline moisture conditions for each period (Van Der Schrier et
al., 2013). This self-calibration enables the scPDSI to more accurately reflect drought trends and
intensities across regions, removing dependency on fixed parameters and baseline periods that
limit the traditional PDSI. In this study, scPDSI data are sourced from the Climatic Research Unit
(CRU), derived from monthly precipitation and temperature fields in the CRU TS 2.1
high-resolution surface climate dataset, covering the period from 1901 to 2022 with a spatial
resolution of 0.5°×0.5°. The scPDSI values typically range from -4 to 4, with more positive
(negative) values representing wetter (drier) conditions.
In addition to the drought index, we used meteorological reanalyses of ERA-5 developed by
the European Centre for Medium-Range Weather Forecasts (ECMWF). ERA-5 represents a major
advance in modeling and data assimilation techniques, combining global observational data with
state-of-the-art model outputs to produce a comprehensive and consistent global dataset (Sun et al.,
2020). This dataset offers hourly estimates at a horizontal resolution of 30-km, with 137 vertical
levels extending from the surface to 80 km in the atmosphere. ERA-5 serves as a crucial resource
for applications in climate research, weather forecasting, environmental monitoring, and
policy-making (Wang et al., 2024; Muñoz-Sabater et al., 2021; Jiao et al., 2021). For this study,
we used ERA-5 surface meteorological variables, including 2-meter air temperature (T2M) and
total precipitation (PRE).





**Table 1** Summary of models participating in the TRENDY project.

| MODEL | CABLE-POP | CLASSIC | CLM5.0 | DLEM | IBIS | JSBACH | ISAM | JULES |
|---|---|---|---|---|---|---|---|---|
| Country | Australia | Canada | USA | USA | China | Germany | USA | UK |
| Resolution | 1°×1° | 1°×1° | 1.25°×1° | 0.5°×0.5° | 1°×1° | 1.875°×1.875° | 0.5°×0.5° | 1.875°×1.25° |
| Carbon–nitrogen interactions | YES | NO | YES | YES | NO | YES | YES | YES |
| Separation of direct and diffuse radiation | YES | NO | YES | NO | NO | NO | NO | YES |
| Reference | (Haverd et al., 2013) | (Melton et al., 2020) | (Lawrence et al., 2019) | (Tian et al., 2015) | (Kucharik et al., 2000) | (Reick et al., 2021) | (Jain and Yang, 2005) | (Clark et al., 2011) |

| MODEL | LPJ | LPJ-GUESS | LPX-Bern | OCN | ORCHIDEE | SDGVM | VISIT | VISIT-NIES | YIBs |
|---|---|---|---|---|---|---|---|---|---|
| Country | USA | Germany | Switzerland | Germany | France | USA | Japan | Japan | China |
| Resolution | 0.5°×0.5° | 0.5°×0.5° | 0.5°×0.5° | 1°×1° | 0.5°×0.5° | 1°×1° | 0.5°×0.5° | 0.5°×0.5° | 1°×1° |
| Carbon–nitrogen interactions | NO | YES | YES | YES | YES | YES | NO | NO | NO |
| Separation of direct and diffuse radiation | NO | NO | NO | NO | NO | NO | NO | NO | YES |
| Reference | (Sitch et al., 2003) | (Smith et al., 2001) | (Spahni et al., 2013) | (Zaehle and Friend, 2010) | (Krinner et al., 2005) | (Walker et al., 2017) | (Kato et al., 2013) | (Ito, 2010) | (Yue and Unger, 2015) |



2.4 Land cover data
The MODIS vegetation cover dataset was used to determine the dominant land cover types
for individual grid cells. This dataset analyzes vegetation reflectance in the visible and
near-infrared wavelengths to produce essential vegetation indices, such as the Normalized
Difference Vegetation Index (NDVI) and EVI (Friedl et al., 2010), offering comprehensive global
coverage with a fine spatial resolution up to 250m and an 8-day temporal interval (Huete et al.,
2002). The MODIS vegetation cover dataset has been widely used in research areas including
ecological monitoring, agricultural and forestry management, climate change studies, and disaster
assessment (Shim et al., 2014; Wu et al., 2019; Wang et al., 2022). For this study, we used MODIS
land cover data version 051, averaged for 2001-2012, to classify dominant vegetation types across
seven categories: $C_3$ grassland (C3G), shrubland (Shr), deciduous broadleaf forest (DBF),
evergreen broadleaf forest (EBF), evergreen needleleaf forest (ENF), cropland (Crop), and $C_4$
grassland (C4G).

2.5 Statistical methods
To explore the relationships between variables, we applied three statistical techniques:
Pearson correlation, partial correlation, and linear regression. The Pearson correlation coefficient γ
was calculated to assess the linear relationship between two variables and is defined as follows:

$$\gamma = \frac{\sum \left(X_i - \overline{X}\right)\left(Y_i - \overline{Y}\right)}{\sqrt{\sum \left(X_i - \overline{X}\right)^2 \sum \left(Y_i - \overline{Y}\right)^2}}$$

where $X_i$ is GPP and $Y_i$ is one of meteorological variables. $\overline{X}$ and $\overline{Y}$ are their mean values.
We conducted a multivariate partial correlation analysis to exclude the potential confounding
effects of other meteorological factors (Zhou et al., 2016; Zhang et al., 2023). For example, the
correlation of $\gamma_{12}$ between GPP and PRE is not only affected by the relationships between GPP
and other meteorological variables, such as T2M ($\gamma_{13}$), but also the relationships between PRE
and T2M ($\gamma_{23}$). As a result, the partial correlation coefficient $\gamma_{12\cdot3}$ is calculated as follows:

$$\gamma_{12\cdot3} = \frac{\gamma_{12} - \gamma_{13}\gamma_{23}}{\sqrt{\left(1 - \gamma_{13}^2\right)\left(1 - \gamma_{23}^2\right)}}$$

This method ensures that the relationship between GPP and PRE is independent of the effects of
T2M, providing a better quantification of their direct associations.
To assess the dependency of GPP on multiple meteorological variables, we applied a multiple
linear regression method. This approach allows us to estimate the individual contributions of each
predictor to GPP while controlling the influence of other factors. The multiple linear regression
model is formulated as:

$$Y = \beta_0 + \beta_1 X_1 + \beta_2 X_2 + \epsilon$$

where Y represents GPP, $X_1$ and $X_2$ denote PRE and T2M respectively. Here, $\beta_0$ is the intercept,
and $\beta_1$, $\beta_2$ are the regression coefficients associated with each predictor. These coefficients
represent the partial effect of each variable on GPP while holding other variables constant. The
term $\epsilon$ accounts for the variability in GPP not explained by the predictors. We assessed the
significance of each coefficient to determine the strength and direction of the relationships
between GPP and individual predictors (PRE or T2M). We also quantify changes in GPP during
years of extreme warming or drought, relative to the mean state. Extreme warming episodes are



identified as years when grid-specific temperatures exceed the 90th percentile, while drought
extremes are defined as years when the scPDSI falls below the 10th percentile. To facilitate the
analyses, we interpolated all datasets, including GPP observations, TRENDY simulations, ERA-5
meteorology, scPDSI, and MODIS land cover, into the same resolution of $1° \times 1°$.

**3 Results**

3.1 Spatiotemporal variations of GPP

We first compared the temporal variations of GPP between observations and simulations (Fig.
1a). Observed GPP from GLASS showed a global mean of 123.73 Pg yr$^{-1}$ and a positive trend of
0.18 Pg yr$^{-2}$ during 1982-2017. Simulated GPP from the S3 run of the TRENDY project exhibited
large inter-model variability, ranging from 95.44 Pg yr$^{-1}$ in CABLE-POP to a maximum of 151.65
Pg yr$^{-1}$ in JULES. Most models predicted positive trends between 0.17 and 0.53 Pg yr$^{-2}$. The
multi-model ensemble (MME) produced an average GPP of 120.06 Pg yr$^{-1}$, closely matching the
GLASS observations. However, the ensemble simulations yielded a positive trend of 0.37 Pg yr$^{-2}$,
almost doubling the estimate with GLASS. In addition to the overall trend, observations showed
substantial year-to-year variations with a standard deviation of 3.16 Pg yr$^{-1}$, while models in
general underestimated this interannual variations, ranging from 1.84 to 5.75 Pg yr$^{-1}$.

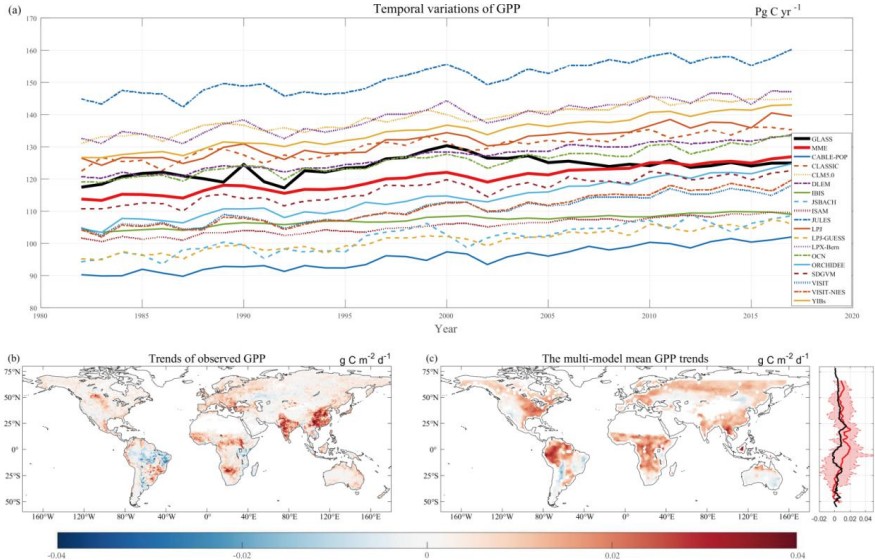

**Fig. 1.** Comparison of spatiotemporal variations in Gross Primary Productivity (GPP) between
observations and model simulations. The (a) temporal variations in simulated GPP from individual
models and the multi-model ensemble (MME) mean (thick red line) are shown alongside
observational data from GLASS (thick black line). Spatial patterns of GPP trends from (b)
observations, represented as the mean of three products (GLASS, GOSIF, JUNG), are compared
with (c) the MME of the simulations. Latitudinal variations in GPP trends are also shown, with
observational data represented in black and model simulations in red; shading indicates one
standard deviation.



The ensemble of three observational datasets revealed large spatial heterogeneity in GPP
trends (Fig. 1b). The greatest GPP enhancement occurred around 25° N, largely driven by hotspots
in eastern China and India. Positive trends were also observed in the Sahel, southern Africa,
Eurasia, and North America. In contrast, GPP in South America, particularly within the Amazon
rainforest, showed a decreasing trend. Similarly, the MME predicted positive GPP trends across
much of the Northern Hemisphere (NH, Fig. 1c), though model simulations underestimated the
GPP increases observed in China and India. Nevertheless, model predictions yielded a notable
positive trend in South America, where observations suggested a decline. Overall, the MME
captured the latitudinal variations in GPP trends but tended to overestimate positive trends in
tropical regions.
3.2 Relationships between GPP and meteorological factors
We analyzed the relationships between GPP and meteorological variables, including
temperature, precipitation, and drought indices (Fig. 2). Observed GPP showed positive responses
to temperature at high latitudes of NH but negative responses at lower latitudes, particularly in the
Southern Hemisphere (SH). Warming promotes GPP at high latitudes by moving temperature
closer to the optimal range for photosynthesis, while inhibiting GPP at lower latitudes where
temperatures often exceed this threshold (Piao et al., 2013). The MME well reproduced this spatial
pattern, reflecting positive correlations in the boreal regions and negative correlations in the
tropics (Fig. 2a). Precipitation was positively correlated with GPP across most tropical and
subtropical areas, especially in arid and semi-arid regions dominated by shallow-rooted vegetation.
In contrast, negative correlations between GPP and precipitation were observed at high latitudes in
the NH (>50° N), where GPP variations depend strongly on solar radiation. Increased rainfall
reduces sunlight availability, inhibiting photosynthesis and resulting in negative correlations
between GPP and precipitation. While the MME generally captured the observed positive
correlations between GPP and precipitation (Fig. 2b), it did not predict the negative correlations
north of 50° N, likely due to an inadequate representation of light dependency in those regions.
GPP was positively correlated with scPDSI in low to middle latitudes but negatively correlated in
boreal regions, mirroring precipitation trends and highlighting precipitation's role in GPP
responses to drought. The MME reproduced these patterns but tended to overestimate positive
correlations, especially in EBF (Fig. 2c).
We then distinguished the relationships between GPP and meteorological factors across
various products/models and vegetation types (Figs. 3 and S1-S3). Observed GPP correlations
with temperature were insignificant for most vegetation types, with the exception of positive
correlations for ENF and negative correlations for C4G (Fig. 3a). The TRENDY models largely
captured these relationships, with 12 out of 17 models yielding positive correlations for ENF and
14 out of 17 showing negative correlations for C4G. Observed correlations with precipitation were
generally positive, especially for C3G and C4G (Fig. 3b). Among the 17 models, 9 predict
significantly positive correlations for both C3G and C4G, suggesting a consistent parameterization
of water stress for these grass species. Positive correlations were more common with drought
indices, particularly for C3G, C4G, Shr, and Crop (Fig. 3c). These patterns were well captured by
the TRENDY models, though they tended to predict positive and mostly significant responses for
EBF, where observed GPP had very low correlations with scPDSI.

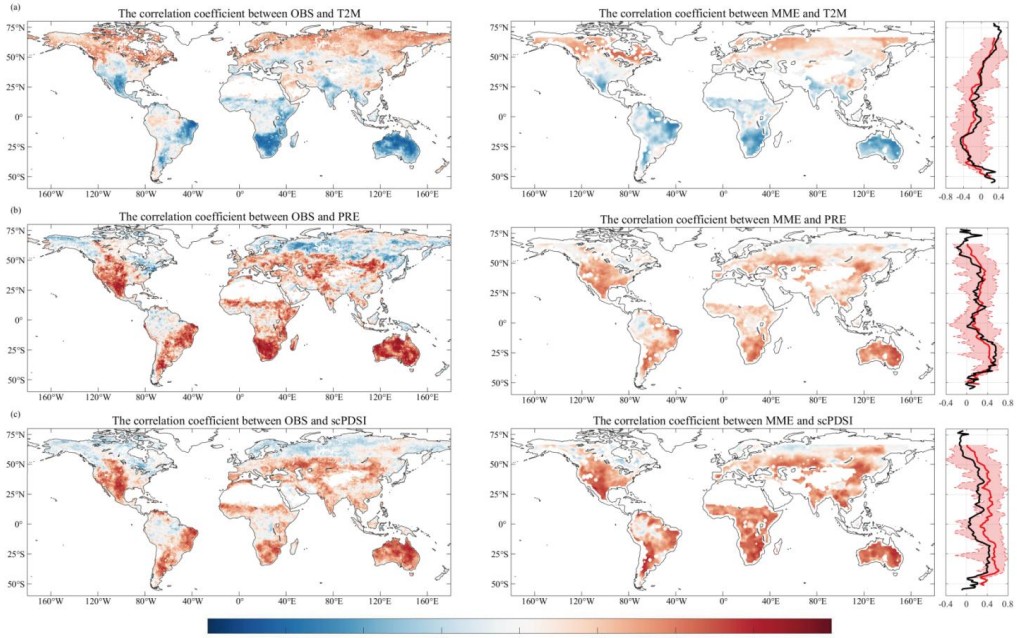

**Fig. 2.** Mean spatial correlation coefficients between GPP and meteorological variables. The
correlation between annual GPP and (a) temperature (T2M), (b) precipitation (PRE), and (c)
self-calibrated Palmer Drought Severity Index (scPDSI) is averaged for observational datasets
(GLASS, GOSIF, JUNG) and multiple model ensemble simulations. For observed GPP,
correlation coefficients were calculated at each grid cell over the period 1982-2017 for GLASS,
2001-2018 for GOSIF, and 1982-2011 for JUNG. For simulated GPP, correlation coefficients were
calculated at each grid cell over the period 1982-2017 for individual models. Latitudinal variations
in correlation coefficients are also shown, with observational data represented in black and model
simulations in red; shading indicates one standard deviation.



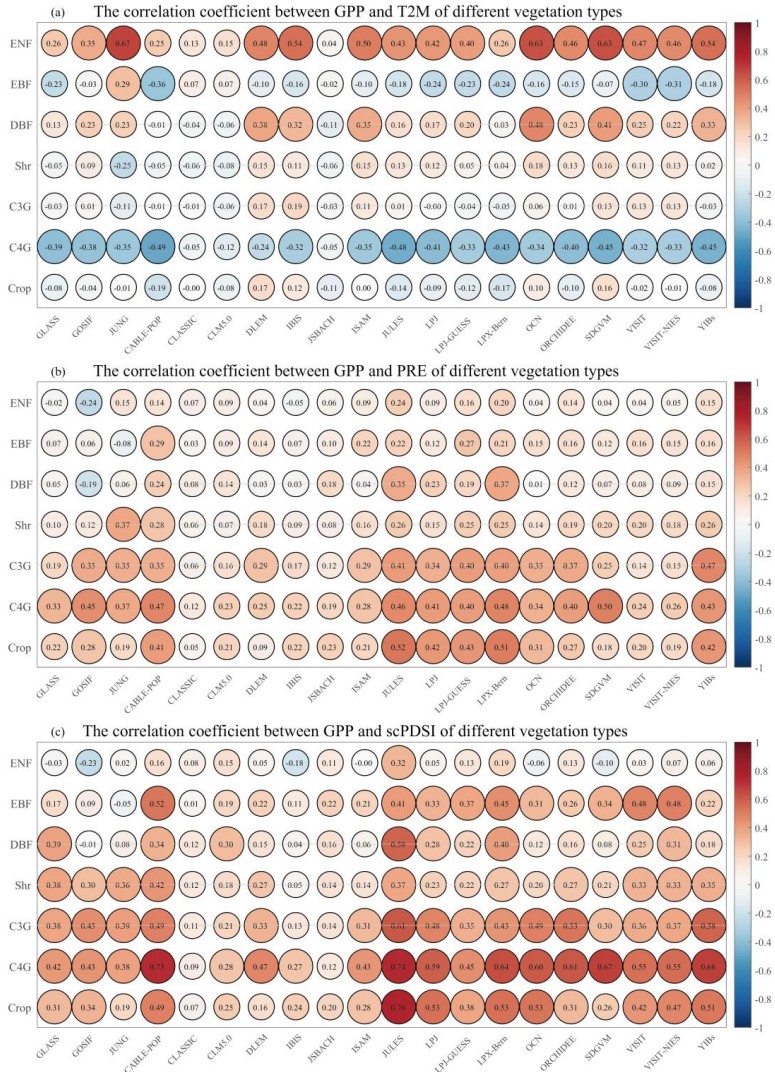

**Fig. 3.** Heatmaps of Pearson correlation coefficients between GPP and meteorological variables across vegetation types. Panels show correlations between GPP and (a) T2M, (b) PRE, and (c) scPDSI for three observational data products (GLASS, GOSIF, and JUNG) and 17 vegetation models. Vegetation types include evergreen needleleaf forest (ENF), deciduous broadleaf forest (DBF), evergreen broadleaf forest (EBF), shrubland (Shr), $C_3$ grassland (C3G), $C_4$ grassland (C4G), and cropland (Crop). Correlation coefficients are represented by circles, with larger circles indicating significant correlations ($P<0.05$) and smaller circles indicating non-significant correlations ($P>0.05$).



We further conducted partial correlation analysis to isolate the independent effects of
individual meteorological variables. For temperature, observed and simulated GPP showed
consistently positive correlations for ENF and DBF, but negative correlations for C4G (Fig. S4a).
For other vegetation types, observed GPP displayed slightly positive but statistically insignificant
correlations, while model simulations generally showed stronger correlations except for EBF,
where negative correlations were achieved. Precipitation exhibited positive correlations with GPP
for C3G, C4G, Shr, and Crop, with stronger correlations in the model simulations compared to
observations (Fig. S4b). For tree species, observations suggested weak correlations between
precipitation and GPP, whereas the models predicted positive correlations. Consequently, the
models showed strong positive GPP responses to scPDSI (Fig. S4c), likely due to the enhanced
amelioration in GPP by precipitation (Fig. S4b). On average, the partial correlation coefficients
between GPP and meteorological variables were 52.02% lower than the direct correlations,
indicating the non-negligible influences of interactions among meteorological factors.

We quantified the sensitivities of GPP to meteorological factors for both observations and
models (Fig. 4). Results showed that one-degree increase in temperature generally led to an
increase of 0.013-0.1g C m$^2$ day$^{-1}$ in GPP across most vegetation types. However, GPP for EBF
and C$_4$ grasslands decreased in response to rising temperatures. Notably, in C$_4$ grasslands,
observed and simulated GPP showed negative responses to temperature, with sensitivities of -0.25
and -0.35 g C m$^2$ day$^{-1}$ K$^{-1}$, respectively (Fig. 4a). Observed GPP showed low sensitivity to
precipitation changes for trees (e.g., DBF, ENF, and EBF), but notably positive responses for
non-tree (e.g., C3G, C4G, shrub, and crop) species (Fig. 4b). Models reproduced the larger
responses of non-tree species compared to trees, though they tended to overestimate the impact of
precipitation on GPP. The sensitivity of GPP to the drought index resembled that to precipitation,
except that observed GPP negatively responded to scPDSI for ENF (Fig. 4c). Simulated GPP
showed positive responses to drought index across all vegetation types, with the strongest
sensitivity for C4G. In summary, the models effectively captured the high sensitivity of tropical
grasslands to drought stress but underestimated the resilience of trees.



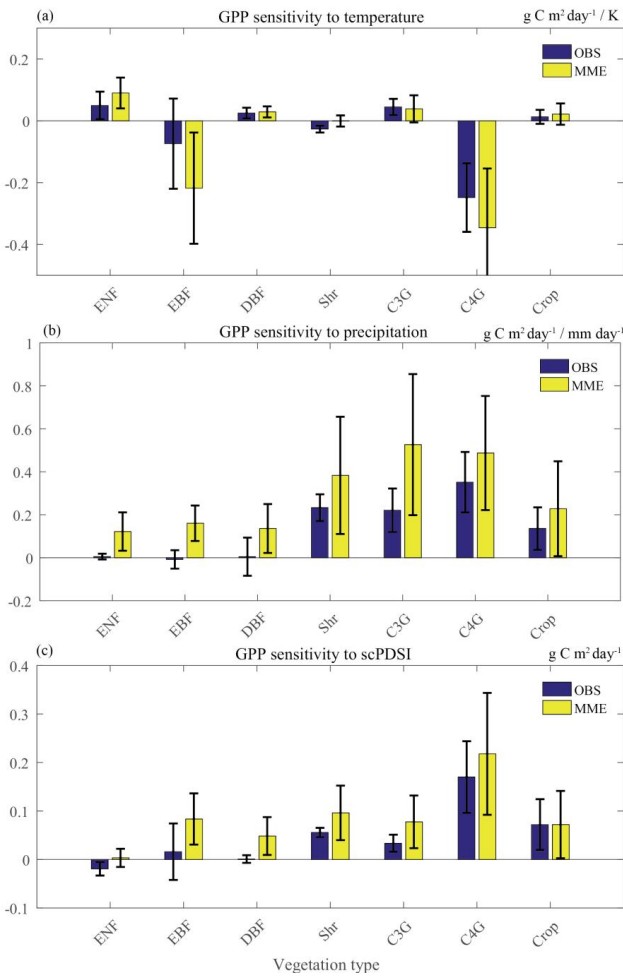


**Fig. 4.** Sensitivities of GPP to meteorological variables across vegetation types. The regression
coefficients are calculated between GPP and (a) T2M, (b) PRE, and (c) scPDSI for both
observations (blue) and model simulations (yellow). The figure presents the mean values from
three observational datasets and multiple models, with errorbars indicating one standard deviation
for either observations or models.


3.3 GPP responses to extreme warming and drought
We examined GPP responses to extreme high temperature and drought conditions (Fig. 5).
Observations showed that extreme high temperatures moderately increased GPP in high latitudes
of the NH, but weakened GPP in low latitudes, particularly in the SH (Fig. 5a). The model
ensemble reproduced this spatial pattern. From a latitudinal perspective, observed and simulated
GPP responses were largely consistent, with limited differences in the magnitude of fluctuations.
Globally, observed GPP generally exhibited a negative response to extreme drought, a pattern that




was well reproduced by the MME. However, the MME tended to overestimate the magnitude of
the negative response compared to observations (Fig. 5b).

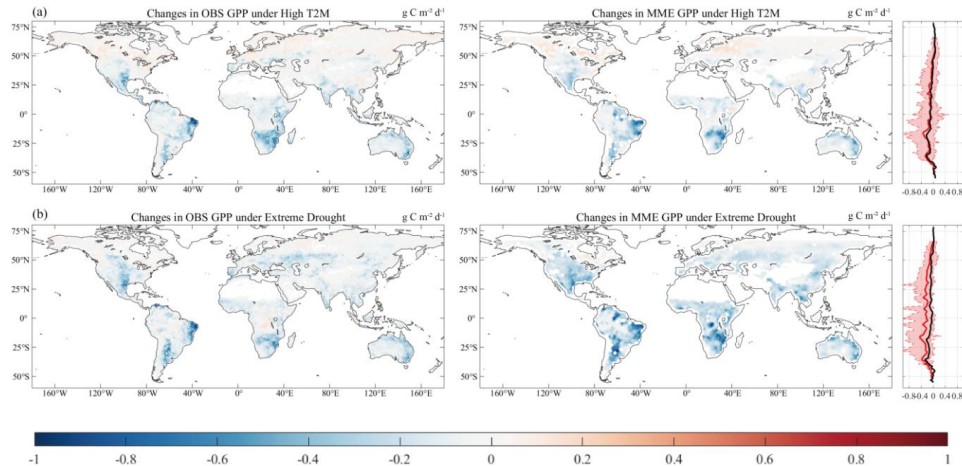


**Fig. 5.** Response of GPP to extreme (a) warming and (b) drought events averaged for observational datasets (GLASS, GOSIF, JUNG) and multiple model ensemble simulations. Changes in GPP are calculated as deviations during years with the (a) highest 10% of temperature or (b) lowest 10% of scPDSI at each grid point, relative to the long-term mean GPP. Latitudinal variations in GPP changes are also shown, with observational data represented in black and model simulations in red; shading indicates one standard deviation.



To better understand these responses, we analyzed GPP responses to extreme warming and

drought across various datasets, models, and vegetation types (Figs S5-S6). For extreme high
temperatures, GPP showed a positive response in DBF and ENF, while other vegetation types
exhibited negative responses. The TRENDY models largely captured these patterns, with only 3
out of 17 models showing a negative response in DBF and all models showing a positive response
in ENF (Fig. S5a). The MME also aligned reasonably well with observed averages, showing
positive responses in DBF and ENF but negative responses in other vegetation types (Fig. S6a).
Under extreme drought conditions, observed GPP consistently exhibited a negative response. This
was well captured by the TRENDY models, though they tended to overestimate the magnitude of
the responses (Fig.S5b). The MME also reproduced the observed negative responses but showed
larger variability among models (Fig. S6b). It should be noted that observed GPP showed positive
feedback to drought in ENF, whereas the MME showed an opposite response.



**4 Discussion**

4.1 Causes of GPP responses to meteorological factors

The response of GPP to temperature varies substantially by latitude, with a stronger positive response in high-latitude regions of the NH and an inverse relationship in low-latitude regions. This pattern is largely driven by the differing sensitivities of vegetation types to temperature changes. In high-latitude areas, which are predominantly covered by coniferous forests, low temperatures are a primary limiting factor for photosynthesis. A rise in temperature can extend the growing season, improve photosynthetic efficiency, and enhance GPP, thereby supporting the growth and carbon absorption capacity of these ecosystems (Sendall et al., 2015; Dusenge et al., 2023; Grossiord et al., 2022). In contrast, low-latitude regions are mainly comprised of EBF and grasslands. In these tropical and subtropical areas, temperatures are usually at or above the optimal thresholds for plant growth. Additional temperature increases can induce heat stress, decrease photosynthesis efficiency, and increase transpiration, leading to water deficits that further inhibit plant growth and GPP (Doughty et al., 2023; Moore et al., 2021).

Increases in precipitation and scPDSI generally indicate improved water availability, which is a fundamental constraint on plant growth, particularly in water-limited regions. The adaptability and physiological characteristics of plants to environmental conditions play a critical role in their responses. When precipitation or drought indices increase, soil moisture conditions improve, enabling plants to perform photosynthesis more effectively (Desai et al., 2022). This leads to an increase in GPP, reflecting enhanced water use efficiency and a growth advantage under adequate water conditions. However, in regions where temperature is not the primary limiting factor for plant growth, higher temperatures can exacerbate transpiration rates, resulting in water deficits (Chen et al., 2023). This dynamic explains why vegetation types that exhibit a negative correlation between GPP and temperature often show a positive correlation between GPP and precipitation or scPDSI (Fig. 2).

4.2 Performance of vegetation models

When evaluating the spatiotemporal variations of GPP, we found that models generally overestimate GPP trends compared to observations, consistent with the findings of Yang et al. (2022). During 1982-2017, the simulated GPP from the TRENDY models exhibited substantial inter-model variability (Fig. 1). Among the models, the correlation between GPP and temperature was accurately captured for most vegetation types (Fig. 3). However, models often overestimated the impact of precipitation on GPP, particularly in vegetation types such as C3G, DBF, ENF, and EBF. Moreover, GPP is typically positively correlated with the drought index, especially for non-tree species such as C3G, C4G, shrubs, and crops. The TRENDY models captured these characteristics well, but they generally overestimated the correlation between scPDSI and GPP. These overestimations are likely due to the improper parameterization of water stress and soil moisture dynamics, which leads to an exaggerated influence of water availability on GPP in the models. The MME helps to constrain inter-model variability and provides a closer estimate of the long-term GPP trend (Fig. 1). However, it still overestimates GPP sensitivities to meteorological variables, including temperature, precipitation, and drought index (Fig.4).

4.3 Uncertainties and implications

There are several uncertainties and limitations in our analyses. First, we utilized



observational datasets from multiple sources, including GLASS, GOSIF, and JUNG, to ensure the
reliability and comprehensiveness of our results. However, these datasets inherently contain
uncertainties, particularly regarding vegetation canopy structure parameters (e.g., leaf area index),
which could affect the derived GPP trends and responses to meteorological variables (Prentice et
al., 2024). Second, we employed multi-model simulations from TRENDY S3 run, which
incorporates interannually varying meteorology and land cover. To exclude the effects of land-use
change, we collected GPP data from the S2 run and re-calculated correlations/regressions with
meteorological variables (Figs. S7-S8). Comparisons showed limited differences between the
results from the S2 and S3 runs, supporting the robustness of our derived climatic impacts on GPP.
Third, the impacts of environmental factors on GPP vary across different time scales (Zhou et al.,
2023). In the short term, fluctuations in climatic factors, such as flash droughts or temperature
variations, can have immediate effects on GPP. In contrast, over longer time scales, factors like
vegetation adaptation, soil moisture dynamics, and long-term climate change may play more
important roles. Neglecting these discrepancies in the time scales may lead to an incomplete
understanding of the driving mechanisms behind GPP variations for both observations and model
simulations.
Despite these limitations, this study systematically quantified the responses of GPP to
different meteorological factors, and compared these responses between observational datasets and
numerical models. It identified the differences in GPP responses to temperature, precipitation, and
drought across various vegetation types, offering helpful insights for improving individual models.
From the perspective of multi-model ensembles, the study assessed the overall performance and
biases of current state-of-the-art vegetation models, as well as their ability to capture GPP
responses to interannual variability and long-term climate change. These findings provide a robust
foundation for understanding multi-model ensemble predictions, particularly in interpretating
long-term trends and interannual fluctuations of terrestrial carbon sinks. This knowledge can
support policymakers and land managers in developing scientifically informed strategies for land
use and ecological conservation, ultimately promoting the resilience of ecosystem to climate
change.
**Data Availability Statement**
The TRENDY v-11 data used for this study could be downloaded from the website
([https://mdosullivan.github.io/GCB/](https://mdosullivan.github.io/GCB/)).
**Author contributions**
XY conceived the project. YZ performed the data processing, conducted the analysis, and wrote
the draft of the paper. XY, XL, and JZ assisted in the interpretation of the results and contributed
to the discussion and improvement of the paper.

**Conflict of Interest**
The authors declare no conflicts of interest.
**Acknowledgements**
We thank all the DGVM authors in TRENDY-v11 project for providing the long-term simulation
data.



**Financial support**

This research was jointly supported by the National Key Research and Development Program of China (grant no. 2023YFF0805402) and the National Natural Science Foundation of China (grant no. 42275128).

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
