# Peer review of "https://doi.org/10.5194/egusphere-2025-1515 Preprint. Discussion started: 24 April 2025 © Author(s) 2025. CC BY 4.0 License."

_EGUsphere, 2025_

## Author Comment (AC1)

We are grateful to the editor and referees for their time and energy in providing helpful comments and guidance that have improved the manuscript. In this document, we describe how we have addressed the reviewers' comments. Please note that the quantified results have slightly changed due to adjustments in the temporal range (1982–2011) and the datasets (specifically, the removal of GOSIF GPP). Referee comments are shown in black italics and author responses are shown in blue regular text. A manuscript with tracking changes is attached at the end.

*Reviewer #1:*

*General comments*

*This manuscript investigates how gross primary productivity (GPP) responds to temperature, precipitation, and drought across global ecosystems, using two satellite-derived GPP products (GLASS and GOSIF), a data-driven GPP product (JUNG) and outputs from 17 DGVMs within the TRENDY v11 multi-model simulations. This study refers to the mean of three GPP products as "observations" and uses them to evaluate the sensitivity of DGVM-simulated GPP to environmental variables in terms of the global spatial pattern and different plant functional types (PFTs). The topic is important for understanding the terrestrial carbon cycle under a changing climate. However, there are several major issues that still need to be addressed regarding terminology, data selection and benchmarking, and incomplete information, all of which undermine the study's reliability. Below, I outline major concerns, missing information, and specific comments for revision.*

➢ Thank you for your evaluations. We have made substantial revisions following your comments. We hope this version of paper have answered your concerns.

*Major concerns*

*The authors refer to GLASS (based on MODIS/AVHRR and LUE models), GOSIF (based on SIF, proxy for GPP), and JUNG (data-driven, upscaled via machine learning) products as "Observations of GPP." This is conceptually inaccurate and misleading. The GLASS and GOSIF GPP products should instead be referred to as "satellite-derived GPP"/ "satellite-based GPP," as none of them are direct observations. I believe this mislabeling may also confuse other readers.*

➢ Thank you for your valuable comment. In the revised manuscript, we have clarified that the GLASS product is satellite-derived and the JUNG product is the machine learning-based upscaled datasets, both of which are not direct

observations. We validated them against FLUXNET eddy covariance tower measurements and found reasonable agreement (see the new Fig. 2). Therefore, we treated them as observation-constrained benchmark datasets for model evaluation and comparison and clarified in the revised paper as follows:

"In this study, we selected 51 eddy covariance sites from a total of 201 FLUXNET sites to evaluate the responses of GPP to temperature and precipitation from the benchmark datasets. The selected sites should meet the following criteria: (1) a record length of at least 8 years, and (2) a missing data ratio of less than 50% for both individual days and years. Although neither the GLASS nor JUNG GPP products represent direct observations, they show good agreement with *in situ* FLUXNET measurements in terms of climatic sensitivity during the specific validations. We therefore consider them reliable observation-constrained benchmark datasets for model evaluation and comparison." (Lines 120-127)

"We evaluated the climatic sensitivity of the benchmark GPP datasets against FLUXNET measurements (Fig. 2). Most FLUXNET sites available for comparison are located in North America and Europe. Benchmark GPP correlated positively with temperature in western Europe, weakly negatively in central Europe, and negatively in southern North America (Fig. 2a). For precipitation, FLUXNET GPP showed positive correlations across Europe, particularly along the Mediterranean and Atlantic coasts, as well as over much of North America (Fig. 2c). The mean of the GLASS and JUNG datasets was reasonably consistent with FLUXNET, with correlation coefficients of 0.63 ($p < 0.01$) for temperature (Fig. 2b) and 0.59 ($p < 0.01$) for precipitation (Fig. 2d)." (Lines 291-299)

[Figure]

**Fig. 2.** Distribution of correlation coefficients between FLUXNET GPP and climatic variables for (a) temperature (T2M) and (c) precipitation (PRE). These correlations are compared with those derived based on benchmark GPP (the mean of GLASS and JUNG) for (b) T2M and (d) PRE at individual sites.

*The authors use the mean of three GPP products as benchmarks to evaluate the sensitivity of TRENDY model simulations to environmental drivers. Although these products have been validated at flux tower sites, their performance is not consistently reliable, especially for long term trends (Zheng et al., 2020, Bai et al., 2023). In addition, Bai et al. (2023) reported large discrepancies in trends among different satellite-derived GPP products. As shown in Figure 4 of this manuscript, for some plant functional types (PFTs), the sensitivity of GPP to climatic variables varies greatly among the three satellite-derived datasets ─ sometimes with differences as large as the sensitivities themselves (e.g., for EBF in Fig. 4a and Fig. 4c). Therefore, it is questionable to use the average of these three products as a robust benchmark.*

➢ Thank you for the valuable comments. To enhance the robustness of our analysis, we have implemented the following improvements in the revised manuscript:

   (1) We removed the GOSIF dataset due to its relatively shorter temporal coverage and retained only the overlapping period (1982-2011) for GLASS and JUNG;

   (2) While all benchmark GPP products include certain uncertainties, both GLASS and JUNG have been validated against FLUXNET site data and demonstrate

consistent responses to key climatic variables. We used the mean of GLASS and JUNG because validations showed that their average performed better compared to individual dataset in capturing the climatic sensitivity from site-level measurement (see the new Fig. S2). However, for most of analyses, we retain the range from the two datasets to quantify the uncertainty.

In the revised paper, we clarified as follows:

"The mean of the GLASS and JUNG datasets was reasonably consistent with FLUXNET, with correlation coefficients of 0.63 ($p < 0.01$) for temperature (Fig. 2b) and 0.59 ($p < 0.01$) for precipitation (Fig. 2d). These correlations are higher than the those derived from individual benchmark GPP datasets (Fig. S2). Therefore, we used the benchmark GPP datasets in subsequent analyses, given the limited spatiotemporal coverage of FLUXNET measurements. To mitigate potential biases while still accounting for uncertainties among individual datasets, we also used their ensemble mean." (Lines 296-302)

[Figure]

**Fig. S2.** Comparison of GPP-climate correlations between FLUXNET *in situ* data and the GLASS or JUNG benchmark datasets for (a) temperature and (b) precipitation. At each FLUXNET site, correlation coefficients are calculated between the FLUXNET eddy covariance GPP and climatic variables, and compared with the corresponding correlations derived from the benchmark datasets.

*The classification of grid cells by plant functional types (PFTs) is based on the 2001 – 2012 MODIS land cover mean, but the analysis period spans as far back as 1982. Since both real-world vegetation and modeled PFTs can shift over time, this temporal mismatch introduces additional uncertainty. A more robust approach would be to focus on flux tower sites with stable vegetation types and use site-level GPP – climate relationships to evaluate both model and satellite-derived GPP responses.*

➢ Thank you for your valuable suggestion. We have added validations of GLASS and JUNG products using FLUXNET flux tower site data in the revised paper. Please see the new Fig. 2 and Fig. S2 in our above responses. We also added a new Fig. S1 to show the distribution of PFTs. We checked that the changes of MODIS PFT from 2001 to 2012 is limited, with the maximum of-2.82% for C4G.

In addition, we compared simulation results from the S2 (without land cover change) and S3 (with land cover change) runs, and found limited differences in the derived GPP sensitivities to climatic drivers from models (see Figs. S6-S7). It suggests that land cover change may not change our main conclusions.

In the revised paper, we have included following discussion on the effect of land cover change: "Second, we employed multi-model simulations from TRENDY S3 run, which incorporates interannually varying meteorology and land cover. To exclude the effects of land-use change, we collected GPP data from the S2 run and re-calculated correlations/regressions with climatic variables (Figs. S6-S7). Comparisons showed limited differences between the results from the S2 and S3 runs, supporting the robustness of our derived climatic impacts on GPP." (Lines 567-571)

[Figure]

**Fig. S1.** Spatial distribution of the dominant land cover type. The number of grids for each plant functional type (PFT) is shown alongside the colorbar. The PFTs include cropland (Crop), $C_4$ grassland (C4G), $C_3$ grassland (C3G), shrubland (Shr), deciduous broadleaf forest (DBF), evergreen needleleaf forest (ENF), and evergreen broadleaf forest (EBF).

*Incomplete information*
*The caption of Figure 2 is the only place where the time periods of the three GPP products are mentioned: "For observed GPP, correlation coefficients were calculated at each grid cell over the period 1982－2017 for GLASS, 2001－2018 for GOSIF, and 1982－2011 for JUNG." The manuscript does not explain how the temporal*

*mismatch was handled, nor how the three datasets were merged. Additionally, it is unclear what time period was used when calculating the correlation between the merged GPP dataset and climatic variables. Please clarify.*

➢ We are sorry about this confusion. In response, we have revised the analysis by removing the GOSIF dataset due to its relatively shorter and misaligned temporal coverage. All analyses now rely solely on the overlapping period (1982–2011) between the GLASS and JUNG datasets. The updated text is as follows:

"In this study, we use the overlapping period of 1982-2011 from both GLASS and JUNG as the reference period." (Lines 127-128)

"To facilitate the analyses, we interpolated all datasets, including GPP benchmark, TRENDY simulations, ERA-5 meteorology, scPDSI, and MODIS land cover, into the same resolution of $1^{o} \times 1^{o}$ using linear interpolation." (Lines 248-251)

*Please provide a supplementary figure showing the spatial distribution and number of grid cells classified under each land cover type.*

➢ Thank you for your suggestion. We have added a new Fig. S1 to display the spatial distribution of dominant plant functional type (PFT) used in our analysis. The number of grids for each PFT is also shown.

[Figure]

**Fig. S1.** Spatial distribution of the dominant land cover type. The number of grids for each plant functional type (PFT) is shown alongside the colorbar. The PFTs include cropland (Crop), $C_4$ grassland (C4G), $C_3$ grassland (C3G), shrubland (Shr), deciduous broadleaf forest (DBF), evergreen needleleaf forest (ENF), and evergreen broadleaf forest (EBF).

*The definition of "sensitivity" is not clearly stated in the manuscript. A formal definition needs to be provided.*

➢ We have added a formal definition of "sensitivity" in the Methods section of the revised manuscript as follows:

 "The multiple linear regression model is formulated as:

$$Y = \beta_0 + \beta_1 X_1 + \beta_2 X_2 + \epsilon \qquad (7)$$

where Y represents GPP, $X_1$ and $X_2$ denote PRE and T2M respectively. Here, $\beta_0$ is the intercept, and $\beta_1$, $\beta_2$ are the regression coefficients associated with each predictor. These coefficients represent the partial effect of each variable on GPP while holding other variables constant, and they are defined as the sensitivities of GPP to the corresponding climatic variables (Piao et al., 2013)." (Lines 235-243)

*In Section 2.3, please include the mathematical formulation of the scPDSI index.*

➢ In the revised manuscript, we have added the detailed mathematical formula of the scPDSI index to Section 2.3 as follows:

"In this study, we used the self-calibrating Palmer Drought Severity Index (scPDSI) to represent drought severity. scPDSI is an improved version of the original PDSI and is widely recognized for its application in drought monitoring and assessment (Wells et al., 2004). The core of PDSI is the water balance equation:

$$P = ET + R + RO + L \qquad (1)$$

where *P* is precipitation, *ET* is evapotranspiration, *R* is soil recharge, *RO* is runoff, and *L* is loss from the soil. These terms are computed using a two-layer soil model (topsoil + underlying layer) and potential values derived from meteorological data (temperature, precipitation, soil water-holding capacity). Meanwhile, the CAFEC precipitation ( $\widehat{P}$ ) is defined as the amount of precipitation needed to maintain "normal" soil moisture conditions:

$$\widehat{P} = \alpha \cdot \widehat{ET} + \beta \cdot \widehat{R} + \gamma \cdot \widehat{RO} + \delta \cdot \widehat{L} \qquad (2)$$

where $\widehat{ET}, \widehat{R}, \widehat{RO}, \widehat{L}$ are the potential values of evapotranspiration, recharge, runoff, and loss, respectively. The coefficients $\alpha$, $\beta$, $\gamma$, $\delta$ are defined as water-balance coefficients, which are set to constant in PDSI calculations. The scPDSI improves the traditional PDSI by incorporating adjusted (self-calibrated) coefficients for each location rather than fixed values. These parameters are derived using extensive historical meteorological data to establish baseline

moisture conditions, improving both the accuracy and stability of the index (Van Der Schrier et al., 2013).

 The departure of actual precipitation from CAFEC precipitation is:

$$d = P - \widehat{P} \tag{3}$$

This departure is scaled by a weighting factor $K$ (also calibrated in scPDSI):

$$Z = K \cdot d \tag{4}$$

where $Z$ is a dimensionless moisture anomaly index for each time step. This $Z$ value is then used to derive scPDSI based on a recursive computation." (Lines 152-182)

*In Section 2.3, please specify the final temporal resolution of the data used in this study. Are the analyses conducted at monthly or annual resolution?*

➢ We clarified that annual mean datasets are used for analyses: "We calculated the annual mean value of both scPDSI and climatic variables and investigated their relationships with GPP from both benchmarks and models." (Lines 199-200)

*The discussion in Section 4.1 is too general —it only addresses GPP responses to temperature, precipitation, and drought on a global scale. Readers would benefit from a more detailed discussion of how these responses differ across vegetation types (especially where GPP responses differ significantly between PFTs).*

➢ Thank you for your constructive suggestion. In the revised manuscript, we have expanded the discussion in Section 4.1 to include a more detailed analysis of how GPP responses to temperature, precipitation, and drought vary across different vegetation types. The changes include:

"In high-latitude areas, which are predominantly covered by coniferous forests and tundra, low temperatures are the primary constraint on photosynthesis. Boreal coniferous forests benefit most from spring warming which extends the photosynthetic period, whereas tundra vegetation shows strongest response to summer warming that accelerates their growth cycles." (Lines 477-480)

"Tropical EBF are particularly sensitive to rising temperature as they already function near their thermal optimum; further warming reduces stomatal conductance and increases photorespiration. Savanna grasslands, although more heat-tolerant, also experience GPP declines under extreme heat, especially when

accompanied by water stress." (Lines 484-487)

"Arid and semi-arid ecosystems, such as dry shrublands, show the strongest GPP responses to precipitation changes, whereas temperate forests exhibit more moderate responses due to lower water limitation." (Lines 495-497)

"Some PFTs show limited GPP responses to climatic variations (Fig. 5), such as DBF and shrubs to temperature, and ENF and EBF to precipitation. This is likely because their GPP are not primarily constrained by these factors under typical environmental conditions. In relatively stable climates, such as temperate forests or humid tropical regions, temperature and water availability often remain near optimal levels for photosynthesis, so additional warming or increased precipitation brings little further benefit and may even reduce GPP (Reichstein et al., 2006). Many of these PFTs also possess structural and physiological traits, such as deep root systems, evergreen leaf phenology, or high thermal tolerance, which buffer them against short-term climate fluctuations and reduce their sensitivity to seasonal or interannual variability in climate (Choat et al., 2018). In some ecosystems, other factors such as light availability, nutrient limitation, or biotic interactions exert stronger control over GPP, further dampening the apparent influence of temperature or precipitation (Nemani et al., 2003)." (Lines 507-518)

*The main text lacks a conclusion section.*

➢ Thank you for your suggestion. In the revised paper, we have added a Conclusion section as follows:

" 5 Conclusion

This study systematically explored the responses of GPP to temperature, precipitation, and scPDSI across various vegetation types. Both the benchmark datasets and MME of simulations showed distinct GPP responses to temperature between boreal and tropical ecosystems, with warming enhancing GPP in high-latitude needleleaf forests but reducing GPP in low-latitude ecosystems such as evergreen broadleaf forests and tropical savannas. Precipitation exhibited positive effects on almost all PFTs, with stronger influences on non-woody vegetation such as C3G, C4G, and croplands. Although current vegetation models generally capture these response patterns, they tend to overestimate the positive effects of precipitation on GPP, particularly in tropical regions. Such biases may lead to an overestimation of carbon sink losses during drought years, as many models fail to adequately represent drought adaptation mechanisms observed in

real ecosystems (Peng et al., 2024). Furthermore, structural differences among models, such as the inclusion of carbon-nitrogen coupling and the separation of direct/diffuse radiation, may substantially affect the simulated GPP sensitivity to drought, highlighting the need for improved process representation in global carbon cycle models." (Lines 599-613)

*Specific comments*

*fig 1a. JUNG and GOSIF GPP time series are missing.*

➤ In the revised version, we have focused solely on GLASS and JUNG datasets as the observation-constrained benchmarks. The time series for JUNG GPP has now been added to Fig. 1a to ensure a complete and accurate representation of the data used in our analysis.

[Figure]

**Fig. 1.** Comparison of spatiotemporal variations in Gross Primary Productivity (GPP) between benchmark and model simulations. The (a) temporal variations in simulated GPP from individual models and the multi-model ensemble (MME) mean (thick red line) are shown alongside benchmark data from GLASS (thick black line) and JUNG (thick blue line). Spatial patterns of GPP trends from (b) benchmark, represented as the mean of two products (GLASS and JUNG), are compared with (c) the MME of the simulations. Latitudinal variations in GPP trends are also shown, with benchmark data represented in black and model simulations in red; shading indicates one standard deviation.

*fig 1. Please Specify the exact time period over which GPP trends were calculated.*

➤ In the revised manuscript, we have clearly specified the time period used for

calculating GPP trends as follows:

"In this study, we use the overlapping period of 1982-2011 from both GLASS and JUNG as the reference period." (Lines 127-128)

"We first compared the temporal variations of GPP between benchmark datasets and simulations during 1982-2011 (Fig. 1a)." (Lines 255-256)

"The two benchmark datasets collectively showed that GPP trends exhibit substantial spatial heterogeneity during 1982-2011 (Fig. 1b)." (Lines 278-299)

*fig 3. Please adding the model ensemble mean result to the figure and analyzing it in the corresponding text.*

➢ We have added the multi-model ensemble mean (MME) to Fig. 4 (the original Fig. 3) as suggested:

[Figure]

**Fig. 4.** Heatmaps of Pearson correlation coefficients between GPP and climatic variables across vegetation types. Panels show correlations between GPP and (a) T2M, (b) PRE, and (c) scPDSI for two benchmark data products (GLASS and JUNG) and 17 vegetation models. Vegetation types include evergreen needleleaf forest (ENF), deciduous broadleaf forest (DBF), evergreen broadleaf forest (EBF), shrubland (Shr), $C_3$ grassland (C3G), $C_4$ grassland (C4G), and cropland (Crop). Correlation coefficients are represented by circles, with larger circles indicating significant correlations ($P<0.05$) and smaller circles indicating non-significant correlations ($P>0.05$).

*fig 4. Suggest adding a global mean bar in this bar plot.*

➢ We have added a global mean bar to Fig. 5 (the original Fig. 4) as suggested:

"Temperature showed moderately positive correlations with both benchmark and simulated GPP on the global scale." (Lines 365-366)

"These negative sensitivities outweighed the positive responses of other vegetation types, leading to an overall negative GPP responses to warming on the global scale." (Lines 381-382)

[Figure]

**Fig. 5.** Correlations and sensitivities of GPP to climatic variables across vegetation types. Panels (a-c) show the partial correlation coefficients between GPP and (a) T2M, (b) PRE, and (c) scPDSI, while panels (d-f) show the corresponding regression coefficients (sensitivities) for (d) T2M, (e) PRE, and (f) scPDSI. Results are presented as the mean across two benchmark datasets (BEN; blue) and the multi-model ensemble (MME; yellow) of S3 runs from 17 models, with errorbars indicating one standard deviation across the benchmark datasets or the models.

*fig 5. The "latitudinal variations" plots lack units.*

➢ The units have been added to the "latitudinal variations" in Fig. 6 (the original Fig. 5) as suggested:

[Figure]

**Fig. 6.** Response of GPP to extreme (a) warming and (b) drought events averaged for benchmark datasets (GLASS and JUNG) and multiple model ensemble simulations. Changes in GPP are calculated as deviations during years with the (a) highest 10% of temperature or (b) lowest 10% of scPDSI at each grid point, relative to the long-term mean GPP. Latitudinal variations in GPP changes are also shown, with benchmark data represented in black and model simulations in red; shading indicates one standard deviation.

*line 22. In sentence: "Precipitation had a relatively low impact on GPP" Please be clear to state relative low positive or negative impact? 一 is this referring to the model or to the GPP products?*

➢ Thank you for your comment. In the revised paper, we clarified as follows:
"Both the benchmark datasets and models indicated a relatively weak but positive effect of precipitation on GPP in deciduous and evergreen forests, whereas non-tree vegetation types, such as grasslands and croplands, showed a much stronger positive response." (Lines 23-26)

*line 61-62. "While this response is protective in the short term, it ultimately leads to a decline in GPP." 一 needs citation.*

➢ Added as suggested:
"While this response is protective in the short term, it ultimately leads to a decline in GPP (Gupta et al., 2020)." (Lines 64-65)

*line 64-68. Only one example is provided. Please add at least one more reference to support the statement: "there has been a notable increase in the sensitivity of global ..."*

➤ Added as suggested:

"Over the past three decades, there has been a notable increase in the sensitivity of global vegetation productivity to drought conditions and this sensitivity has maintained an upward trend. For example, Wei et al. (2023) found that the sensitivity of GPP to drought rose by 13.76% in 2006-2018 compared to 1993-2005. Chen et al. (2025) projected that under future climate scenarios, extreme droughts will increasingly impact GPP, especially in semi-arid zones (drought index 0.15-0.8)." (Lines 68-73)

*line 81-83. Also needs at least one more reference to support the statement: "the need of careful calibration and validation using observed data to improve model reliability."*

➤ Added as suggested:

"This underscores the need of careful calibration and validation using observed data to improve model reliability (Zheng et al., 2020)." (Lines 89-90)

*line 101. LAI is not an "environmental factor," but rather a vegetation structural parameter. Please revise.*

➤ Corrected as suggested:

"It integrates data from the Moderate Resolution Imaging Spectroradiometer (MODIS) and Advanced Very High Resolution Radiometer (AVHRR), using optimized light-use efficiency models to estimate GPP by combining absorbed PAR with vegetation structural parameter (e.g., leaf area index) and environmental factors (e.g., shortwave radiation) (Liang et al., 2024)." (Lines 105-109)

*line 101-103. Since the study analyzes GLASS GPP responses and uses GLASS as a benchmark, please also cite literature showing the consistency between long-term GLASS GPP and tower-based observations.*

➤ In the revised manuscript, we have added citations to literature that demonstrates the consistency between long-term GLASS GPP products and tower-based observations:

"The GLASS dataset has been validated against ground measurements and aligns well with tower-based observations, capturing seasonal and interannual variability across ecosystems (Ma and Liang, 2022; Bai et al., 2023)." (Lines 109-111)

*line 136. It is unclear whether the paper uses S2 or S3 TRENDY simulations—please clarify.*

➢ We have clarified in the revised manuscript that this study uses the S3 simulations from the TRENDY project. The updated text now explicitly states:

"For this study, we analyzed simulated GPP data from the 17 DGVMs for the S3 experiments, focusing on GPP responses to changes in major climatic variables, and used the same method to analyze the S2 experiment for validation." (Lines 144-146)

*line 204-205. The interpolation method is not described—please add.*

➢ The interpolation method has been clearly described in the revised manuscript. The added text is as follows:

"To facilitate the analyses, we interpolated all datasets, including GPP benchmark, TRENDY simulations, ERA-5 meteorology, scPDSI, and MODIS land cover, into the same resolution of $1^{\circ} \times 1^{\circ}$ using linear interpolation." (Lines 248-251)

*line 213. "Most models predicted..." — please specify the exact number of models (e.g., X out of 17).*

➢ The statement has been revised to specify the exact range of model predictions. The updated text now is as follows:

"The global GPP trend predicted by these models is between 0.17 and 0.51 Pg C $yr^{-2}$." (Lines 260-261)

In the following analyses, we specified the number of models that are classified into a group:

"The TRENDY models largely captured these relationships, with 13 out of 17 models yielding significantly positive correlations for ENF and 13 out of 17 showing significantly negative correlations for C4G." (Lines 345-347)

"Among the 17 models, 10 predict significantly positive correlations for both C3G and C4G, suggesting a consistent parameterization of water stress for these grass species." (Lines 345-350)

*line 229. The phrase "The ensemble of three observational datasets revealed large spatial heterogeneity in GPP trends" is ambiguous — it could be interpreted as inconsistency among datasets. If the intended meaning is that GPP trends themselves are spatially variable, please reword.*

➢ We have revised the text as follows:

"The two benchmark datasets collectively showed that GPP trends exhibit substantial spatial heterogeneity during 1982-2011 (Fig. 1b)." (Lines 278-279)

*line 236-203. The statement that "Overall, the MME captured the latitudinal variations in GPP trends but tended to overestimate positive trends in tropical regions." is inaccurate. According to Figure 1, the trends in tropical regions differ in sign between MME and satellite products. Please revise.*

➢ Thank you for your comments. The statement has been revised in the manuscript to accurately reflect the discrepancy between the MME and satellite products in tropical regions. The updated text now is as follows:

"Overall, the MME captured the latitudinal variations in GPP trends, but showed a GPP trend pattern that contrasts sharply with benchmark in tropical regions, particularly in the Amazon region where the model predicted growth while benchmark showed a decline." (Lines 286-288)

*line 255. The phrase "likely due to an inadequate representation of light dependency" requires a citation.*

➢ Added as suggested:

"While the MME generally captured the observed positive correlations between GPP and precipitation (Fig. 3b), it did not predict the negative correlations north of $50^o$ N, likely due to an inadequate representation of light dependency in those regions (Pierrat et al., 2022)." (Lines 322-325)

*line 261. "(Figs. 3 and S1－S3)" should be "Figures S1－S3."*

➢ Corrected as suggested.

*line 296. "Tree species" is first defined in Line 308. Please move or adjust for clarity.*

➢ This sentence has been deleted in the revised manuscript.

*line 289-301. Figure S4 is cited four times in the main text. If it is so central to the*

*analysis, consider moving it into the main text.*

➢ In the revised paper, we have merged the original Fig. S4 with the original Fig. 4 to form a new Fig. 5 in the main text.

[Figure]

**Fig. 5.** Correlations and sensitivities of GPP to climatic variables across vegetation types. Panels (a-c) show the partial correlation coefficients between GPP and (a) T2M, (b) PRE, and (c) scPDSI, while panels (d-f) show the corresponding regression coefficients (sensitivities) for (d) T2M, (e) PRE, and (f) scPDSI. Results are presented as the mean across two benchmark datasets (BEN; blue) and the multi-model ensemble (MME; yellow) of S3 runs from 17 models, with errorbars indicating one standard deviation across the benchmark datasets or the models.

*line 305. "in C4 grasslands" should be "for C4 grasslands."*

➢ Corrected as suggested.

*line 303-305. GPP from shrublands also decreases with rising temperature ─this information is missing.*

➢ Yes. The GPP of shrubs decreases with rising temperature. However, its magnitude is much lower compared to EBF and C4 grassland. In the revised paper, we clarified as follows:

"However, GPP for EBF and C4G largely decreased in response to rising temperatures." (Line 379)

*line 343. "various dataset" is undefined - please clarify.*

➢ We clarified as follows:

"To better understand these responses, we analyzed GPP responses to extreme warming and drought across benchmark datasets, models, and vegetation types." (Lines 419-420)

*line 342-353. Same issue as above with Line 289 – 301 — avoid repeatedly citing supplementary figures in the main text. Either integrate them or move the relevant discussion to the supplement.*

➢ We have moved the original Figs S5 and S6 into the main text as the new Figs 7 and 8.

[Figure]

**Fig. 7.** Heatmaps of GPP responses to extreme (a) warming and (b) drought across different vegetation types. Changes in GPP are shown for years with the top 10% of (a) highest temperatures and (b) lowest scPDSI values, relative to the mean state, based on two benchmark datasets (GLASS and JUNG) and 17 vegetation models. The MME of model simulations is also presented.

[Figure]

**Fig. 8.** GPP responses to extreme warming and drought conditions across vegetation types. Differences in GPP are calculated as deviations during years with the top 10% (a) highest temperature or (b) lowest scPDSI values, relative to the long-term mean, for individual grids. These differences are aggregated by seven vegetation types and averaged for two benchmark datasets (blue) and the multi-model ensemble mean (yellow). The errorbars indicate one standard deviation across benchmark datasets or models.

*line 359. In addition to coniferous forests, high-latitude regions also include tundra, deciduous broadleaf forests, and wetlands, etc. Please revise.*

➤ The description has been revised in the manuscript to more accurately represent the ecosystem diversity in high-latitude regions:

"In high-latitude areas, which are predominantly covered by coniferous forests and tundra, low temperatures are the primary constraint on photosynthesis."

(Lines 477-478)

*line 390. The phrase "improper parameterization" is too vague. It sounds like the models are fundamentally flawed. Please revise or provide a specific reference.*

➢ We revised the text as follows:

"The discrepancies suggest that current parameterizations of water stress and soil moisture dynamics could overestimate the sensitivity of GPP to water availability." (Lines 539-540)

*line 420. The manuscript does not analyze interannual variability of GPP response to climatic variables. Please revise the statement accordingly.*

➢ We revised the text as follows:

"From the perspective of multi-model ensembles, the study assessed the overall performance and biases of current state-of-the-art vegetation models, as well as their ability to capture GPP responses to long-term climate change." (Lines 591-593)

**References**

Bai, Y., Liang, S., Jia, A., and Li, S.: Different Satellite Products Revealing Variable Trends in Global Gross Primary Production, Journal of Geophysical Research: Biogeosciences, 128, https://doi.org/10.1029/2022jg006918, 2023.

Chen, Z., Qian, Z., Huang, B., Feng, G., and Sun, G.: Increased Drought Impacts on Vegetation Productivity in Drylands Under Climate Change, Geophysical Research Letters, 52, https://doi.org/10.1029/2025gl115616, 2025.

Choat, B., Brodribb, T. J., Brodersen, C. R., Duursma, R. A., López, R., and Medlyn, B. E.: Triggers of tree mortality under drought, Nature, 558, 531-539, https://doi.org/10.1038/s41586-018-0240-x, 2018.

Gupta, A., Rico-Medina, A., and Caño-Delgado, A. I.: The physiology of plant responses to drought, Science, 368, 266-269, https://doi.org/10.1126/science.aaz7614, 2020.

Liang, S., He, T., Huang, J., Jia, A., Zhang, Y., Cao, Y., Chen, X., Chen, X., Cheng, J., Jiang, B., Jin, H., Li, A., Li, S., Li, X., Liu, L., Liu, X., Ma, H., Ma, Y., Song, D.-X., Sun, L., Yao, Y., Yuan, W., Zhang, G., Zhang, Y., and Song, L.: Advancements in high-resolution land surface satellite products: A comprehensive review of inversion algorithms, products and challenges, Science of Remote Sensing, 10, https://doi.org/10.1016/j.srs.2024.100152, 2024.

Ma, H. and Liang, S.: Development of the GLASS 250-m leaf area index product (version 6) from MODIS data using the bidirectional LSTM deep learning model, Remote Sensing of Environment, 273, https://doi.org/10.1016/j.rse.2022.112985, 2022.

Nemani, R. R., Keeling, C. D., Hashimoto, H., Jolly, W. M., Piper, S. C., Tucker, C. J., Myneni, R. B., and Running, S. W.: Climate-Driven Increases in Global Terrestrial Net Primary Production from 1982 to 1999, Science, 300, 1560-1563, https://doi.org/doi:10.1126/science.1082750, 2003.

Piao, S., Sitch, S., Ciais, P., Friedlingstein, P., Peylin, P., Wang, X., Ahlström, A., Anav, A., Canadell, J. G., Cong, N., Huntingford, C., Jung, M., Levis, S., Levy, P. E., Li, J., Lin, X., Lomas, M. R., Lu, M., Luo, Y., Ma, Y., Myneni, R. B., Poulter, B., Sun, Z., Wang, T., Viovy, N., Zaehle, S., and Zeng, N.: Evaluation of terrestrial carbon cycle models for their response to climate variability and to $CO_2$ trends, Global Change Biology, 19, 2117-2132, https://doi.org/10.1111/gcb.12187, 2013.

Pierrat, Z., Magney, T., Parazoo, N. C., Grossmann, K., Bowling, D. R., Seibt, U., Johnson, B., Helgason, W., Barr, A., Bortnik, J., Norton, A., Maguire, A., Frankenberg, C., and Stutz, J.: Diurnal and Seasonal Dynamics of Solar-Induced Chlorophyll Fluorescence, Vegetation Indices, and Gross Primary Productivity in the Boreal Forest, Journal of Geophysical Research: Biogeosciences, 127, https://doi.org/10.1029/2021jg006588, 2022.

Reichstein, M., Ciais, P., Papale, D., Valentini, R., Running, S., Viovy, N., Cramer, W., Granier, A., OgÉE, J., Allard, V., Aubinet, M., Bernhofer, C., Buchmann, N., Carrara, A., GrÜNwald, T., Heimann, M., Heinesch, B., Knohl, A., Kutsch, W., Loustau, D., Manca, G., Matteucci, G., Miglietta, F., Ourcival, J. M., Pilegaard, K., Pumpanen, J., Rambal, S., Schaphoff, S., Seufert, G., Soussana, J. F., Sanz, M. J., Vesala, T., and Zhao, M.: Reduction of ecosystem productivity and respiration during the European summer 2003 climate anomaly: a joint flux tower, remote sensing and modelling analysis, Global Change Biology, 13, 634-651, https://doi.org/10.1111/j.1365-2486.2006.01224.x, 2006.

van der Schrier, G., Barichivich, J., Briffa, K. R., and Jones, P. D.: A scPDSI-based global data set of dry and wet spells for 1901–2009, Journal of Geophysical Research: Atmospheres, 118, 4025-4048, https://doi.org/10.1002/jgrd.50355, 2013.

Wei, X., He, W., Zhou, Y., Cheng, N., Xiao, J., Bi, W., Liu, Y., Sun, S., and Ju, W.: Increased Sensitivity of Global Vegetation Productivity to Drought Over the Recent Three Decades, Journal of Geophysical Research: Atmospheres, 128, https://doi.org/10.1029/2022jd037504, 2023.

Wells, N., Goddard, S., and Hayes, M. J.: A Self-Calibrating Palmer Drought Severity Index, Journal of Climate, 17, 2335-2351, https://doi.org/10.1175/1520-0442(2004)017<2335:ASPDSI>2.0.CO;2, 2004.

Zheng, Y., Shen, R., Wang, Y., Li, X., Liu, S., Liang, S., Chen, J. M., Ju, W., Zhang, L., and Yuan, W.: Improved estimate of global gross primary production for reproducing its long-term variation, 1982–2017, Earth System Science Data, 12, 2725-2746, https://doi.org/10.5194/essd-12-2725-2020, 2020.

---

## Author Comment (AC2)

We are grateful to the editor and referees for their time and energy in providing helpful comments and guidance that have improved the manuscript. In this document, we describe how we have addressed the reviewers' comments. Please note that the quantified results have slightly changed due to adjustments in the temporal range (1982–2011) and the datasets (specifically, the removal of GOSIF GPP). Referee comments are shown in black italics and author responses are shown in blue regular text. A manuscript with tracking changes is attached at the end.

*Reviewer #2:*

*General comments*

*The authors utilized existing GPP observations and multi-model simulations to examine latitudinal differences in temperature responses (positive in boreal, negative in tropics) and vegetation-type specific sensitivities. While the topic is within the scope of Biogeosciences, I have several major concerns regarding the novelty and methodology of this work.*

➢ Thank you for your evaluations. We made substantial revisions following your comments. We hope this version of paper have answered your concerns.

*Numerous previous studies have already investigated GPP responses to temperature, precipitation, and drought across different regions and vegetation types, as well as global GPP responses to specific meteorological factors, i.e., tropical region (Piao et al., 2013, Lomax et al. 2024), boreal forest to drought (Lindroth et al. 2020, Martínez-García et al., 2024). The general relationships between GPP and meteorological variables described in the abstract could essentially be obtained through literature review alone. Therefore, I am not fully convinced by the motivation, novelty, and critical insights in re-examining these well-documented responses.*

➢ Thank you for your insightful comment. We acknowledge that previous studies have extensively investigated the responses of GPP to climatic drivers. However, we provided several new insights as listed below:

(1) This study systematically explored the responses of GPP to temperature, precipitation, and drought across various vegetation types. Most previous studies have focused on either a single vegetation type (e.g., boreal forests in Lindroth et al., 2020) or a single climatic factor (e.g., precipitation in Lomax et al., 2024).

(2) We validated the performance of state-of-the-art vegetation models, which provide long-term carbon simulations for the annual Global Carbon Budget

report, against available benchmark datasets. This updated assessment (e.g., compared with Piao et al., 2013) offers an up-to-date understanding of the credibility of the land carbon budget predicted by these models.

(3) Our analyses showed that state-of-the-art models tend to overestimate the positive effects of precipitation on GPP, particularly in tropical regions. Such biases may lead to an overestimation of carbon sink losses during drought years. Furthermore, structural differences among models, such as the inclusion of carbon-nitrogen coupling and the separation of direct and diffuse radiation, can substantially influence the simulated GPP sensitivity to drought, offering critical insights for model improvement.

In this revised paper, we have incorporated site-level validations of benchmark GPP (Fig. 2), analyzed the simulated GPP sensitivity to climatic factors across different model structures (Fig. 9), and substantially expanded the discussion to examine the drivers of GPP responses and the sources of biases in model simulations. Detailed revisions will be shown in the following responses. With these updates, along with the original analyses, we aim to provide a robust foundation for understanding multi-model ensemble predictions, particularly for interpreting long-term trends and interannual fluctuations of terrestrial carbon sinks.

*The authors acknowledged the potential nonlinear relationship between GPP and meteorological variables in the introduction, yet the analysis relied entirely on linear regression/correlation methods. This approach is inadequate for capturing non-linear GPP responses to extremes (e.g., drought thresholds, temperature optima).*

➢ Thank you for your insightful comment. During the revision process, we attempted to perform quadratic regressions to derive the optimal thresholds as suggested. However, these regressions yielded low $R^2$ values for most plant functional types (not shown), indicating that the nonlinear relationships between GPP and meteorological variables are highly complex and cannot be adequately captured by a simple threshold. Moreover, for some PFTs, such as evergreen trees in boreal regions, environmental temperatures are generally too low to reach their optimal levels, making it difficult to determine such thresholds.

In this revision, we have expanded the discussion to acknowledge the importance

of the nonlinear responses of GPP to climatic factors:

"Some PFTs show limited GPP responses to climatic variations (Fig. 5), such as DBF and shrubs to temperature, and ENF and EBF to precipitation. This is likely because their GPP are not primarily constrained by these factors under typical environmental conditions. In relatively stable climates, such as temperate forests or humid tropical regions, temperature and water availability often remain near optimal levels for photosynthesis, so additional warming or increased precipitation brings little further benefit and may even reduce GPP(Reichstein et al., 2006)." (Lines 507-512)

We also explicitly discussed the limitations of ignoring nonlinear responses:

"Finally, the nonlinear effects of climatic factors on GPP responses were not considered in the analyses. We employed the linear regression to estimate GPP sensitivity to the climatic changes. In reality, GPP often exhibits threshold-like responses to climate drivers, such as declining photosynthetic efficiency under heat stress (Doughty et al., 2023) or diminishing GPP with increasing precipitation once water availability is no longer limiting (Li et al., 2022). Ignoring these nonlinear dynamics may lead to underestimation or overestimation of GPP sensitivities, particularly in regions where climatic conditions approach environmental limits for plant growth. Future studies could incorporate nonlinear analyzing approaches, such as quadratic or piecewise regressions, to better capture the full range of GPP responses to climate variability." (Lines 577-586)

*Additionally, the observational datasets (GLASS, GOSIF, JUNG) cover different periods (1982 – 2017 vs. 2001 – 2018 vs. 1982 – 2011). The non-overlapping timeframes may introduce biases in trend and sensitivity analyses, particularly given accelerated climate change after the 2000s.*

➢ Thank you for your insightful comment. We have revised the analysis by removing the GOSIF dataset due to its relatively shorter and misaligned temporal coverage. All analyses now rely on the overlapping period (1982 – 2011) between the GLASS and JUNG datasets. The updated text is as follows:

"In this study, we use the overlapping period of 1982-2011 from both GLASS and JUNG as the reference period." (Lines 127-128)

*The 17 models included in this study vary substantially in resolution, carbon-nitrogen coupling, and radiation schemes as illustrated in Table 1. The authors should address*

*how structural differences contribute to inter-model variability through sensitivity analyses or other methods.*

➢ Thank you for your valuable suggestion. In the revised paper, we conducted a sensitivity analysis to assess the impact of model structure on the simulated GPP responses to climatic drivers. We added a new section 3.5 and Fig. 9 as follows:

"3.5 Impact of model structure on drought sensitivity

The TRENDY models use different structures and parameterizations for carbon cycle simulations (Table 1), leading to varied GPP responses to climatic variables (Fig. 4). To assess the impact of model structure on simulated GPP sensitivity to drought, we grouped the models based on whether they include carbon-nitrogen coupling and/or whether they use separate schemes for diffuse and direct radiation. For most PFTs, incorporating the nitrogen cycle generally increased GPP sensitivity to scPDSI compared to models without carbon-nitrogen coupling, leading to a further overestimation of drought responses relative to the benchmark datasets (Fig. 9a). Meanwhile, models that distinguish between direct and diffuse radiation showed higher GPP sensitivity to scPDSI than those without such a radiation scheme on the global scale (Fig. 9b). This feature is most pronounced for non-tree PFTs (e.g., C3G, C4G, and crop). In contrast to the effect of carbon-nitrogen coupling, which consistently magnified the simulated drought responses, the two-leaf radiation scheme reduced GPP sensitivity to scPDSI for EBF and shrubs, thereby improving the simulated drought responses for these two PFTs." (Lines 449-462)

[Figure]

**Fig. 9.** Comparison of GPP sensitivity to scPDSI across model configurations. Model groups are categorized based on the presence or absence of carbon-nitrogen (C-N) coupling and the separate treatment of diffuse and direct radiation (two-leaf). For each structural feature, ensemble means are calculated from benchmark datasets (grey; including GLASS and JUNG), models with the feature (blue), and models without the feature (pink). The errorbars indicate one standard deviation across benchmark datasets or model simulations.

We also expanded discussion for the possible causes and implications:

"Our analyses showed that differences in model structures partly contribute to the variations in simulated GPP sensitivities (Fig. 9). For instance, the representations of carbon-nitrogen (C-N) coupling can substantially influence the simulated response to water stress (Luo et al., 2008; Liu et al., 2025). By emphasizing nitrogen limitation on photosynthesis, C-N coupled models may amplify the interactions between water availability and carbon cycling in nitrogen-limited

ecosystems such as ENF, potentially leading to deviations from observed sensitivity patterns (Yang et al., 2025). Furthermore, differences in canopy radiative scheme affect simulated GPP responses to drought. In ecosystems with complex canopy structures, such as EBF and shrubs, models incorporating a two-leaf radiation transfer scheme show better agreement with observed GPP sensitivity to scPDSI. This likely reflects a more realistic representation of light distribution and leaf functional traits in multi-layered canopies, enhancing diffuse radiation use efficiency and mitigating the limiting effect of water stress on photosynthesis (De Pury and Farquhar, 2008). In contrast, in grassland ecosystems with simplified canopy structure (both $C_3$ and $C_4$), models that separate direct and diffuse radiation tend to overestimate GPP sensitivity to scPDSI, likely due to overestimated diffuse fertilization effects (Kanniah et al., 2013)." (Lines 544-558)

*Specific comments*

*Please clarify all the units, Pg C or Pg CO2, throughout the manuscript.*

➢ We clarified that the units of "Pg C" was used throughout the manuscript. All relevant figures, tables, and text have been updated accordingly.

*Only the GLASS GPP product is shown in Figure 1a, while the other two observation datasets are not included. Please explain the rationale for this selective presentation.*

➢ In the revised version, we have focused on GLASS and JUNG datasets as the observational benchmarks. The time series for JUNG GPP has now been added to Fig. 1a to ensure a complete and accurate representation of the data used in our analysis.

[Figure]

**Fig. 1.** Comparison of spatiotemporal variations in Gross Primary Productivity (GPP) between benchmark and model simulations. The (a) temporal variations in simulated GPP from individual models and the multi-model ensemble (MME) mean (thick red line) are shown alongside benchmark data from GLASS (thick black line) and JUNG (thick blue line). Spatial patterns of GPP trends from (b) benchmark, represented as the mean of two products (GLASS and JUNG), are compared with (c) the MME of the simulations. Latitudinal variations in GPP trends are also shown, with benchmark data represented in black and model simulations in red; shading indicates one standard deviation.

*Line 380-383: The divergence between observations and models is already evident in the GPP trends shown in Figure 1. This discrepancy should be explained before analyzing the GPP responses to meteorological factors.*

➢ Section 4.1 focuses on the drivers of GPP sensitivity to climatic factors based on benchmark products, while Section 4.2 addresses the performance of model simulations. Accordingly, we explain the divergence between benchmark data and model results at the beginning of Section 4.2 as follows:

"For this study, we used two benchmark datasets to assess the simulated GPP responses to climatic variables from 17 state-of-the-art vegetation models. The model in general captured the increasing trends in benchmark GPP but with large inter-model variability (Fig. 1). Meanwhile, the MME failed to reproduce the decline in GPP over in Amazon. These biases likely originate from the inadequate

representation of key processes in the models, such as moisture stress, phenological responses, and anthropogenic disturbances (e.g., deforestation and forest degradation) in tropical ecosystems (Gu et al., 2002; Koch et al., 2021). Many models also overestimate the benefits of $CO_2$ fertilization under moisture-constrained conditions, while failing to adequately represent the constraints imposed by combined heat and drought stress on photosynthetic activity, such as photoinhibition and stomatal closure (Green et al., 2020). Furthermore, insufficient representation of thermal adaptation and resilience mechanisms in tropical vegetation may lead to an overestimation of carbon sink capacity under prolonged climatic stress (Doughty et al., 2023)." (Lines 521-532)

*Line 385-388: references are needed for this. Is this from your analysis or previous studies? Please clarify and provide appropriate citations.*

➢ This statement is based on our analyses, particularly the results shown in Fig. 5. In the revised manuscript, we have clarified it as follows:

"However, our analyses showed that models overestimated the impact of precipitation on GPP, particularly in vegetation types such as C3G, DBF, ENF, and EBF (Fig. 5)." (Lines 534-536)

**References**

De Pury, D. G. G. and Farquhar, G. D.: Simple scaling of photosynthesis from leaves to canopies without the errors of big-leaf models, Plant, Cell & Environment, 20, 537-557, https://doi.org/10.1111/j.1365-3040.1997.00094.x, 2008.

Doughty, C. E., Keany, J. M., Wiebe, B. C., Rey-Sanchez, C., Carter, K. R., Middleby, K. B., Cheesman, A. W., Goulden, M. L., da Rocha, H. R., Miller, S. D., Malhi, Y., Fauset, S., Gloor, E., Slot, M., Oliveras Menor, I., Crous, K. Y., Goldsmith, G. R., and Fisher, J. B.: Tropical forests are approaching critical temperature thresholds, Nature, 621, 105-111, https://doi.org/10.1038/s41586-023-06391-z, 2023.

Green, J. K., Berry, J., Ciais, P., Zhang, Y., and Gentine, P.: Amazon rainforest photosynthesis increases in response to atmospheric dryness, Sci Adv, 6, https://doi.org/10.1126/sciadv.abb7232, 2020.

Gu, L., Baldocchi, D., Verma, S. B., Black, T. A., Vesala, T., Falge, E. M., and Dowty, P. R.: Advantages of diffuse radiation for terrestrial ecosystem productivity, Journal of Geophysical Research: Atmospheres, 107, https://doi.org/10.1029/2001jd001242, 2002.

Kanniah, K. D., Beringer, J., and Hutley, L.: Exploring the link between clouds, radiation, and canopy productivity of tropical savannas, Agricultural and Forest Meteorology, 182-183, 304-313, https://doi.org/10.1016/j.agrformet.2013.06.010, 2013.

Koch, A., Hubau, W., and Lewis, S. L.: Earth System Models Are Not Capturing Present-Day Tropical Forest Carbon Dynamics, Earth's Future, 9, https://doi.org/10.1029/2020ef001874, 2021.

Li, W., Migliavacca, M., Forkel, M., Denissen, J. M. C., Reichstein, M., Yang, H., Duveiller, G., Weber, U., and Orth, R.: Widespread increasing vegetation sensitivity to soil moisture, Nature Communications, 13, https://doi.org/10.1038/s41467-022-31667-9, 2022.

Liu, H., Gao, X., Fan, W., and Fu, X.: Optimizing carbon and nitrogen metabolism in plants: From fundamental principles to practical applications, J Integr Plant Biol, 67, 1447-1466, https://doi.org/10.1111/jipb.13919, 2025.

Luo, Y., Gerten, D., Le Maire, G., Parton, W. J., Weng, E., Zhou, X., Keough, C., Beier, C., Ciais, P., Cramer, W., Dukes, J. S., Emmett, B., Hanson, P. J., Knapp, A., Linder, S., Nepstad, D. A. N., and Rustad, L.: Modeled interactive effects of precipitation, temperature, and [CO2] on ecosystem carbon and water dynamics in different climatic zones, Global Change Biology, 14, 1986-1999, https://doi.org/10.1111/j.1365-2486.2008.01629.x, 2008.

Reichstein, M., Ciais, P., Papale, D., Valentini, R., Running, S., Viovy, N., Cramer, W., Granier, A., OgÉE, J., Allard, V., Aubinet, M., Bernhofer, C., Buchmann, N., Carrara, A., GrÜNwald, T., Heimann, M., Heinesch, B., Knohl, A., Kutsch, W., Loustau, D., Manca, G., Matteucci, G., Miglietta, F., Ourcival, J. M., Pilegaard, K., Pumpanen, J., Rambal, S., Schaphoff, S., Seufert, G., Soussana, J. F., Sanz, M. J., Vesala, T., and Zhao, M.: Reduction of ecosystem productivity and respiration during the European summer 2003 climate anomaly: a joint flux tower, remote sensing and modelling analysis, Global Change Biology, 13, 634-651, https://doi.org/10.1111/j.1365-2486.2006.01224.x, 2006.

Yang, H., Zhang, P., Wang, G., Wang, Q., Wang, D., Wang, R., Zhang, X., and Yin, H.: The synergistic strategy of leaf nitrogen conservation and root nitrogen acquisition in an alpine coniferous forest along an elevation gradient, Plant and Soil, https://doi.org/10.1007/s11104-025-07514-3, 2025.